# A Novel ST-ViBe Algorithm for Satellite Fog Detection at Dawn and Dusk

**Huiyun Ma** [1], **Zengwei Liu** [1], **Kun Jiang** [1], **Bingbo Jiang** [2], **Huihui Feng** [1,*] and **Shuaifeng Hu** [1]

1   School of Geosciences and Info-Physics, Central South University, Changsha 410083, China
2   PowerChina Zhongnan Engineering Corporation Limited, Changsha 410014, China
*   Correspondence: hhfeng@csu.edu.cn

**Abstract:** Satellite remote sensing provides a potential technology for detecting fog at dawn and dusk on a large scale. However, the spectral characteristics of fog at dawn and dusk are similar to those of the ground surface, which makes satellite-based fog detection difficult. With the aid of time-series datasets from the Himawari-8 (H8)/AHI, this study proposed a novel algorithm of the self-adaptive threshold of visual background extractor (ST-ViBe) model for satellite fog detection at dawn and dusk. Methodologically, the background model was first built using the difference between MIR and TIR (BTD) and the local binary similarity patterns (LBSP) operator. Second, BTD and scale invariant local ternary pattern (SILTP) texture features were coupled to form scene factors, and the detection threshold of each pixel was determined adaptively to eliminate the influence of the solar zenith angles. The background model was updated rapidly by accelerating the updating rate and increasing the updating quantity. Finally, the residual clouds were removed with the traditional cloud removal method to achieve accurate detection of fog at dawn and dusk over a large area. The validation results demonstrated that the ST-ViBe algorithm could detect fog at dawn and dusk precisely, and on a large scale. The probability of detection, false alarm ratio, and critical success index were 72.5%, 18.5%, 62.4% at dawn (8:00) and 70.6%, 33.6%, 52.3% at dusk (17:00), respectively. Meanwhile, the algorithm mitigated the limitations of the traditional algorithms, such as illumination mutation, missing detection, and residual shadow. The results of this study could guide satellite fog detection at dawn and dusk and improve the detection of similar targets.

**Keywords:** fog detection; dawn and dusk; ViBe; adaptive threshold; H8/AHI

## 1. Introduction

As a common weather phenomenon, fog tends to occur during traffic peaks at dawn and dusk, which has a significant adverse impact on both respiratory health and traffic safety, and it has become an important priority for monitoring by meteorological and environmental departments [1–3]. Traditional fog detection relies on ground observations, which have difficulty reflecting the formation and evolution of fog in a large area [4]. With the rapid development of meteorological satellites, remote sensing technology has been widely used in fog detection due to its large observation range, high temporal resolution and low cost [1,5–7].

Generally, remote sensing-based fog detection relies on the difference in radiation and texture characteristics in visible and infrared bands between background information, such as ground surface and fog [8,9]. For example, the reflectivity of day fog is smaller than that of medium-high clouds but higher than that of water and land surfaces. Meanwhile, fog's top texture is smooth and uniform, while the medium-high cloud is rough [10]. Night fog detection is mainly based on its emission characteristics. Hunt [11] theoretically proved that the emissivity of MIR in the range of 3.5–3.8 μm is less than the emissivity of TIR in the range of 8.5–13 μm. Based on this feature, the brightness temperature difference (BTD) between MIR and TIR is effective for night fog detection [12]. However, relevant studies

still face great challenges in fog detection at dawn and dusk, mainly because of the high solar zenith angles, and the small differences in temperature and spectral characteristics between fog and background [13,14]. The above problems make it difficult for visible, near-infrared, or thermal infrared remote sensing to carry out accurate detection of dawn/dusk fog effectively. In contrast, the radiation energy of the MIR band not only comes from the target's own thermal radiation but also includes the reflection of solar radiation energy. The radiation characteristics vary greatly with the solar zenith angles, resulting in significant changes in the fog BTD with the solar zenith angles [15], while the surface BTD remains relatively constant. The BTD time-series dynamic characteristics of fog and land surface lay a physical foundation for remote sensing-based fog detection at dawn and dusk. The third-generation geostationary meteorological satellite Himawari-8 (H8) with high time resolution (10 min) is one of the few current satellite platforms that can capture the dynamic changes in fog radiation over a large range [16]. The time-series datasets generated by Himawari-8/Advanced Himawari Imager (H8/AHI) can provide the data basis for the dawn/dusk fog detection algorithm based on the time-series image. It is hypothesized to be capable of achieving fog detection at high solar zenith angles. However, there are few relevant studies at high solar zenith angles [17,18].

With the development of computer image processing technology, video moving target detection algorithms are becoming increasingly mature. The visual background extractor (ViBe) algorithm performs information extraction based on object motion and spectral change characteristics, which can accurately capture changing targets in complex scenes and can provide technical support for the specific implementation of fog detection algorithms [19]. However, the traditional ViBe algorithm still faces many limitations in detecting fog at dawn and dusk: (1) the traditional ViBe algorithm mostly builds the background model based on the gray value of the image and fails to make full use of the difference in textural features between the fog and other backgrounds, resulting in the difficulty of separating clouds and fog; (2) being affected by factors such as solar zenith angles, even the same image, BTD has spatial heterogeneity, and the single threshold used in the traditional ViBe algorithm leads to both missed detection and false detection; and (3) the traditional ViBe algorithm mostly uses high-frequency data for foreground target detection, while the monitoring frequency of existing satellites is limited, and the target separation accuracy will be greatly affected due to slow background updates.

To solve the above problems, this study proposes a novel algorithm of the self-adaptive threshold of visual background extractor (ST-ViBe) model for satellite fog detection at dawn and dusk through the time series H8/AHI datasets. By adding the neighborhood mean and variance in local binary similarity patterns (LBSP) in the BTD image, the background model was established, and the texture information was incorporated into the algorithm to improve the accuracy of cloud separation. By combining the BTD and scale invariant local ternary patterns (SILTP), the scene factor was constructed to determine the foreground detection threshold of each pixel, and the local adaptive threshold of the large-range ViBe was established. At the same time, the background model was rapidly updated by accelerating the updating rate and increasing the updating quantity to solve the problem of the limited monitoring frequency of remote sensing images. This study is expected to provide a simple and feasible algorithm for fog detection at dawn and dusk, aiding both regional fog monitoring and forecasting.

## 2. Materials and Methods

### 2.1. Study Region and Datasets

#### 2.1.1. Study Area

The land area of central and eastern China, which spans 104.5°E to 135°E longitude and 33°N to 47.5°N latitude, was selected as the study area. The study area is close to the Bohai Sea and the Yellow Sea, having abundant water vapor. Therein are found the first and second divisions of China's topography, the Northeast Plain and the North China Plain, which are prone to water vapor accumulation. Affected by the temperate monsoon climate,

the area experiences continuous foggy weather in winter and spring [20,21]. Additionally, the area has a relatively developed economy and a large population, and the frequent occurrence of fog has a significant adverse effect on local economic development and human health.

### 2.1.2. Datasets and Preprocessing

Several datasets were required for carrying out this study, which are described as follows:

#### Satellite Datasets

Himawari-8 satellite observation data were selected for fog detection because it is one of the few current satellite data platforms that can capture the dynamic changes in fog over a large range [16]. The dataset had a spatial resolution of 2 km and a temporal resolution of 10 min. Five band datasets of visible (*R*, 0.64 μm), mid-infrared (MIR, 3.9 μm), and thermal infrared (TIR, 8.6; 10.4 and 11.2 μm) were adopted at 06:30–09:00 (dawn) and 15:30–18:00 (dusk), Beijing Time (8 h after the Universal Time Coordinated (UTC)).

#### Validation Datasets

Ground observation datasets from the National Meteorological Information Center of China were used to execute and validate the algorithm proposed in this study. Because of the different monitoring frequencies and space distribution of the observation sites in study area, different sites were adopted at 8:00 and 17:00 after strict quality control (Figure 1). The dataset was described in detail in the study of Chen and Wang [20].

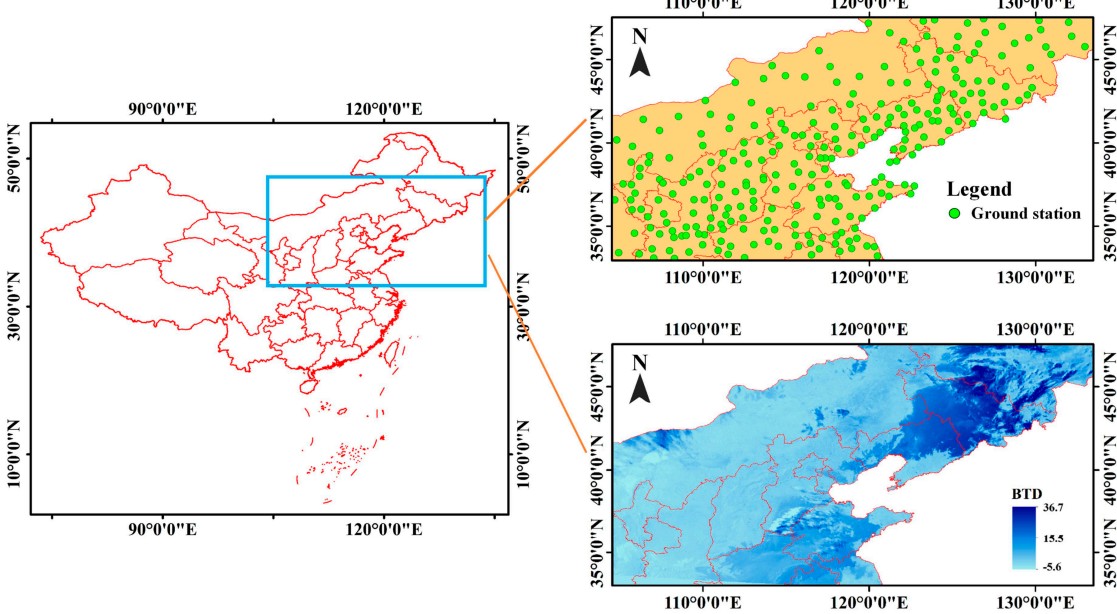

**Figure 1.** Study area distribution of ground observation stations and Bright temperature difference image (Band7 (3.9 μm)−Band14 (11.2 μm)). The image was taken at 8:00 on 30 November 2015.

We evaluated the performance of our algorithm on several randomly selected days and across different conditions. Specifically, data sets from 26 November to 30 November in 2015 were adopted to evaluate the performance of fog detection over a long duration. Furthermore, we randomly selected a daily data set in each month of 2017 to validate the algorithm over different seasons. Fog is usually defined as a weather condition with a visibility of less than 1 km, which can be further divided into sub-categories of strong (visibility is between 0 m and 200 m), dense (200 m to 500 m), and haze (500 m to 1000 m) [22]. However, the definition of haze varies in different regions. According to the Grade of Fog

Forecast promulgated by China Meteorological Administration (CMA) (*GB/T 27964-2011 GB/T*), haze is defined as the horizontal visibility ranges from 1.0 to 10.0 km [23]. By considering the criteria above, this study identified haze as matching the ground station codes 40–49 (fog occurs during observation) or 10 (haze occurs during observation), wherein the visibility is between 500 m and 10 km.

## 2.2. Physical Basis

To accurately detect fog at dawn and dusk, this study was carried out mainly based on the radiation variation characteristics of fog and the land surface under different solar zenith angles. Specifically, night fog only emits in MIR and TIR, the former being usually smaller than the latter, and its BTD is less than 0 K (the blue rectangle in Figure 2a,b) In addition to emissions, fog also reflects MIR radiation in the daytime. In addition, both the brightness temperature ($BT_{MIR}$) and BTD increase with decreasing solar zenith angles. The solar zenith angles are big near the terminator line, and the BTD of fog is close to (or slightly larger than) that of the surface. The closer to the daytime area, the smaller the solar zenith angles, and the fog BTD becomes gradually larger than the surface BTD (the red and green rectangles in Figure 2a,b). The above variations in fog radiation characteristics have laid a physical foundation for fog detection at dawn and dusk.

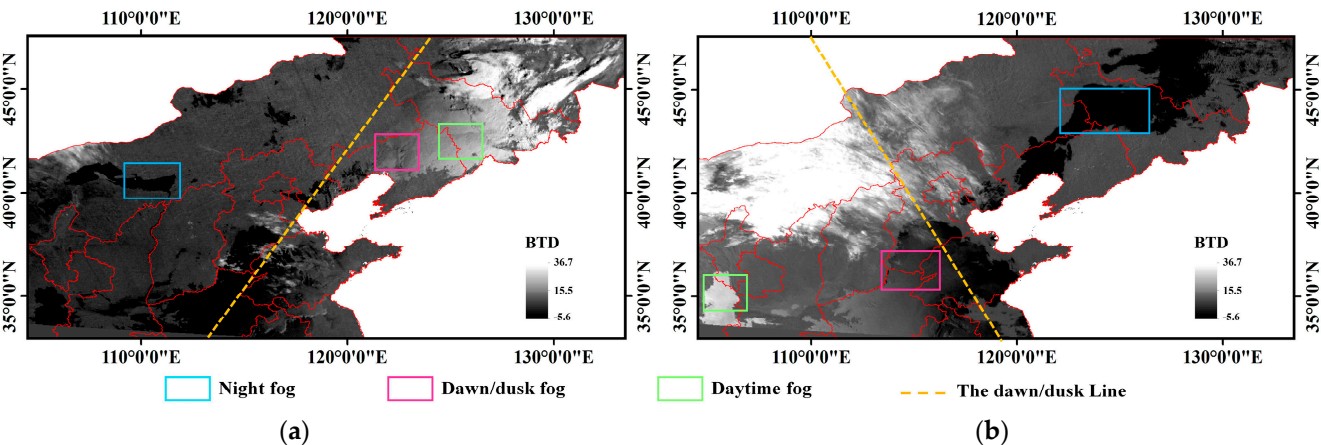

**Figure 2.** Bright temperature difference images at (**a**) 7:30, and (**b**) 16:40, on 30 November 2015.

Figure 3a,b further show the variation characteristics of fog and land surface BTD at dawn and dusk. The emissivity of fog in MIR was less than that of TIR before 7:10, and its corresponding BTD was lower than 0, while the emissivity of the surface in MIR was close to that of TIR, and its BTD was close to 0. At dawn (after 7:10), the reflection of the solar radiation in MIR by the fog increased gradually, and the BTD gradually increased until 7:40, approaching that of the surface. With the gradual decrease in the solar zenith angles (7:50−9:00), the reflectance of fog in MIR was greater than that of the surface, and its BTD growth rate was likewise greater than that of the surface (Figure 3c). The BTD trend of dusk line transit was opposite to that of dawn line transit (Figure 3b). The difference between fog and surface radiation changes at dawn and dusk provided the physical basis for fog detection based on time-series images at dawn and dusk; that is, BTD that varied greatly was the foreground target (fog), and the other becomes the background. Figure 3c,d further quantify the change rate of fog and surface BTD at dawn and dusk near the terminator. The change rate of surface BTD at dawn (0.3–1.5) was usually greater than that of dusk (0–0.6).

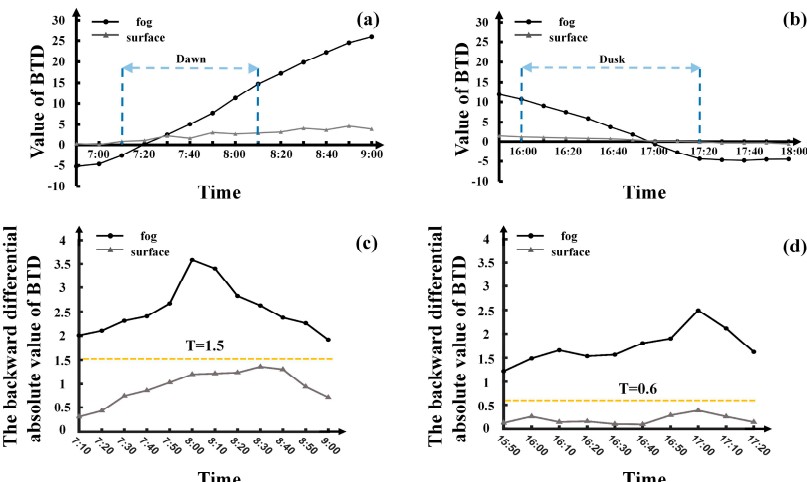

**Figure 3.** BTD of fog and surface at (**a**) dawn, and (**b**) dusk. The backward differential absolute value of fog and surface brightness temperature difference at (**c**) dawn, and (**d**) dusk (i.e., the absolute values of BTD at the measurement time minus the BTD at the previous time). These data were recorded in Zhengzhou city, Henan Province, from 6:30 to 18:00 on 29 November 2015. All the recorded times are in Beijing Time (8 h after the UTC).

### 2.3. ST-ViBe Algorithm Principle

Based on the BTD variation characteristics of fog and surface at dawn and dusk, this study carried out remote-sensing-based fog detection based on the ViBe algorithm. This algorithm is a background modeling method proposed by Olivier Barnich and Marc Van Droogenbroeck [24], the main strategy of which was to randomly collect samples from the area surrounding the current pixel to serve as background model elements based on the random idea. During target detection, the current pixel was then compared with its background model, and if it matched its conditions, it was judged as a background point; otherwise, it was deemed a foreground point. Finally, the background model of the current pixel was then updated randomly, according to the classification results. The process of the algorithm is as follows:

Background model initialization

Select the initial image as the sample to establish the background and build the background model sample set $BM(I)$ for each pixel $I$:

$$BM(I) = \{I_1, I_2, \dots I_n\} \tag{1}$$

#### 2.3.1. Target Detection

Calculate the difference between the background pixel brightness temperature difference $I_t$ and the current pixel brightness temperature difference $I_c$ in background model $BM(I)$. Count the number of the absolute values of differences less than threshold R($N_{intersect}$). When $N_{intersect}$ exceeds the minimum intersection number *Min* (Formulas (2) and (3)), the pixel is considered a background pixel (*BG*), and the background model sampling set $BM(I)$ is updated. Otherwise, the pixel is considered foreground (*FD*), and the background model is not updated.

$$N_{intersect} = \sum_{i=1}^{n} (if(|I_t - I_c| < R)) \tag{2}$$

$$I = \begin{cases} BG & N_{intersect} \geq Min \\ FD & N_{intersect} < Min \end{cases} \tag{3}$$

#### 2.3.2. Update of the Background Model

After pixel $I_t$ is judged as the background, the background model $BM(I)$ needs to be updated to ensure the real-time performance of the background detection. The ViBe

algorithm adopts a memory-free update strategy, and the update frequency is $\varphi$; that is, $I_t$ has a probability of $1/\varphi$ to update its corresponding background model. During the update, $I_t$ is used to replace a certain $I_i$ in $n$ samples of the background model, any pixel in the eight neighborhoods of $I_t$ is selected to participate in the update of the background model, and the pixel value of $I_t$ is also used to complete the update.

### 2.4. ST-ViBe Model Construction for Dawn and Dusk Fog Detection

The ViBe algorithm can reduce both the process of background model building and the influence of background light on achieving fog extraction, but it can easily form a shadow area [25]. In addition, ViBe uses a single threshold to extract the foreground in the whole image, which may cause missed detection and false detection, and the background update efficiency is low. Aiming at addressing the above problems, this study innovatively constructed a self-adaptive threshold ViBe (ST-ViBe) fog detection algorithm by coupling LBSP and the SILTP operator. The main innovations of the algorithm were as follows (Figure 4): (1) the LBSP operator was used to increase the neighborhood mean and variance in BTD image to establish the background model and enhance the discrimination between cloud and fog; (2) scene factors were generated by coupling pixel BTD and SILTP texture features. The threshold of the foreground detection distance measure was adjusted adaptively according to the scene factors to remove the problem of large foreground BTD changes caused by the solar zenith angles. (3) By accelerating the background update rate and increasing the number of background updates, background updates were carried out quickly to improve the accuracy of target location detection. (4) Ice clouds, thin clouds, and low clouds in the image were removed with the traditional cloud removal algorithm, residual shadows with multiple images were removed, and the integrity of the detection result was improved by removing small patches and smoothing.

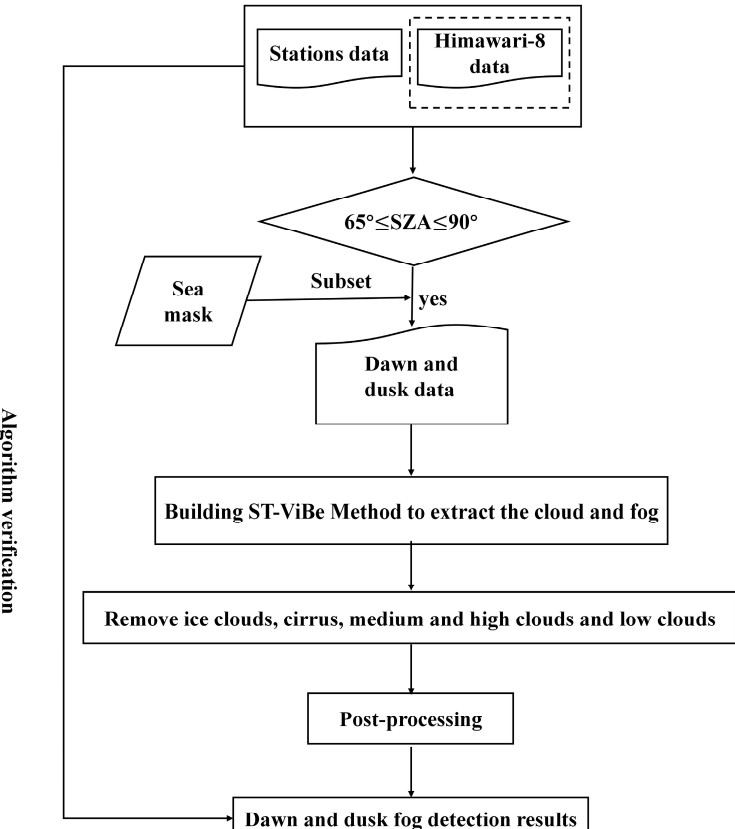

**Figure 4.** Flowchart of the ST-ViBe fog detection algorithm.

The specific steps of the algorithm are as follows:

### 2.4.1. ST-ViBe Background Model Initialization

The LBSP operator was used to separate clouds and fogs when the texture features of clouds and fogs differed greatly. This operator is a texture feature descriptor designed by Bilodeau G A et al. in 2013 [26]. LBSP has high feature description accuracy by introducing a local similarity threshold, comparing the similarity of each pixel with its 16 neighbor pixels, and then carrying out binary coding [27]. In this study, a $5 \times 5$ window (Figure 5) was selected to establish two background models—$BM(I)_1$ and $BM(I)_2$—of the mean variance in BTD between the central pixel and its LBSP neighborhoods, where $BM(I)_1 = \{I_1, I_2, \ldots I_n\}$, with $I_t$ representing the BTD of the neighboring field pixel, and $BM(I)_2 = \{(m_1, s_1), (m_2, s_2), \ldots (m_n, s_n)\}$, where $(m_t, s_t)$ is the BTD mean and variance in the LBSP operator (Figure 5) in the neighboring field pixel. The sample mean ($\overline{m}_M$) and variance ($\overline{s}_M$) in the background model $BM(I)_2$ were initialized to determine the initial values of parameters *Min* and *R* [28], reduce the probability that clouds with complex textures were detected as fog, and improve foreground and background detection accuracy.

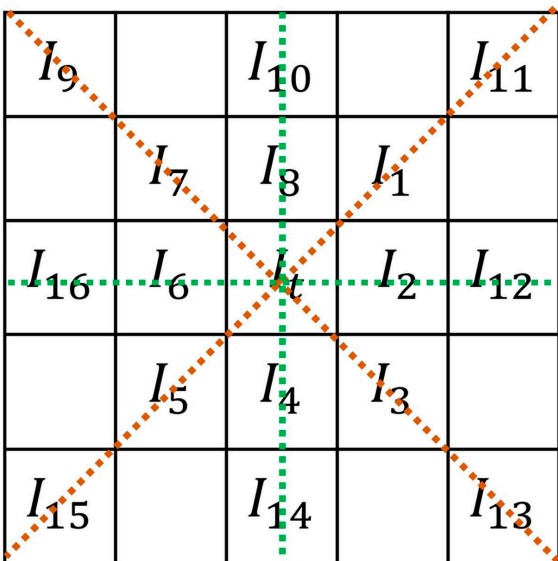

**Figure 5.** LBSP coding neighborhood field.

### 2.4.2. Establishing the Foreground Detection Parameter Set

Traditional ViBe uses a single threshold to detect the foreground; however, the fog occurrence range is usually large with strong spectral spatial heterogeneity, leading to large errors in fog detection. In this study, coupled with the pixel brightness temperature difference, SILTP texture feature coding was introduced to construct scene factor *L*. The Foreground detection parameter (FDPS($x$,$y$)) was constructed together with background models $BM(I)_1$ and $BM(I)_2$ to adaptively determine the distance measurement threshold *R* for foreground detection. SILTP texture feature coding was proposed by Liao S. et al. [28]. By improving the coding method of the operator, the adaptability of texture features to image scale changes was enhanced, thereby effectively reducing the influence of illumination. Additionally, the SILTP operator has strong stability under proportional increases in illumination and sudden changes in local illumination and noise, and can maintain good grayscale invariance, both of which are suitable for judging pixel texture in scenes with large illumination changes [29,30]. The SILTP texture feature coding of pixel $I_{(x,y)}$ was calculated as follows:

$$s_\tau(I_c, I_k) = \begin{cases} 01, & I_k > (1+\tau)I_c \\ 10, & I_k < (1-\tau)I_c \\ 00, & \text{other} \end{cases} \tag{4}$$

$$SILTP(x,y) = \oplus_{k=1}^{8} s_\tau(I_c, I_k) \tag{5}$$

where $I_c$ is the BTD value of pixel $I_{(x,y)}$. $I_k$ is the BTD value in the eight neighborhoods of pixel $I_{(x,y)}$. $\tau$ is the allowable fluctuation parameter; $\tau = 0.3$, according to the spectral variation characteristics of the target in this study. $s_\tau(I_c, I_k)$ is the coding factor of the grayscale difference between the neighboring pixel and the central pixel $I_{(x,y)}$. $\oplus$ represents the hexadecimal binary code $SILTP(x, y)$ formed by the order $s_\tau(I_c, I_k)$.

Defining the parameter $NUM\_SILTP_{(x,y)}$ as the number of 1 occurrence in the $SILTP(x,y)$ code (between 0 and 8), the fog edge and the ground $NUM\_SILTP_{(x,y)}$ is between 5 and 8 when the solar zenith angles is larger than 90°. The ground and interior of fog $NUM\_SILTP_{(x,y)}$ is between 0 and 2 when the solar zenith angles are small. The scene factor $L$ was constructed by coupling the pixel brightness temperature difference and the SILTP texture feature, and the calculation method is shown in Formula (6).

$$L_c = I_c / NUM\_SILTP_{(x,y)} \tag{6}$$

The pixel foreground detection parameters set $FDPS(x, y) = \left\{ I_c, m_c, s_c, NUM\_SILTP_{(x,y)}, L_c \right\}$ were composed of pixel BTD ($I_c$), LBSP neighborhood brightness temperature difference mean and variance ($m_c, s_c$), SILTP texture feature ($NUM\_SILTP_{(x,y)}$), and scene factor ($L_c$).

2.4.3. Determination of the Initial Value of the Model Parameters and Adaptive Adjustment of the Distance Measurement Threshold

The initial values of the ST-ViBe model parameters have a great influence on the results. Fog detection accuracy can be improved by reasonably determining the initial values of the model parameters according to the changes in the brightness temperature differences in fog between dawn and dusk. According to the variation in fog brightness temperature difference between dawn and dusk, an iterative algorithm was used to determine its key parameter values, among which the background sample parameter $n = 20$, the initial value of the minimum intersection number $Min = 4$, and the initial threshold value of distance measurement $R = 3$ K.

Determine the distance measurement threshold $R$ according to the foreground detection parameter set $FDPS(x, y)$. The specific steps are as follows:

(1) Use the initialized background model to initially adjust the parameters $Min$ and $R$ [28]. Calculate the sample mean ($\overline{m}_M$) and variance ($\overline{s}_M$) in the background model $BM(I)_2$. If the current frame pixel $I_c$ is within $[\overline{m}_M - 2\overline{s}_M, \overline{m}_M + 2\overline{s}_M]$ (indicating that the pixel has a high probability of being the background), reduce the minimum intersection number $Min$, increase the distance measure threshold $R$, and increase the possibility of its detection as the background:

$$\begin{cases} Min = Min - 1; \ R = R \times 4; & I_c \in (\overline{m}_M - 2\overline{s}_M, \overline{m}_M + 2\overline{s}_M) \\ Min = Min; \ R = R; & I_c \notin (\overline{m}_M - 2\overline{s}_M, \overline{m}_M + 2\overline{s}_M) \end{cases} \tag{7}$$

(2) According to the scene factor $L$, the distance measurement threshold $R$ is further adjusted for the two scenes, dawn and dusk.

The method for determining $R$ at dawn is:

$$R_{dawn} = \begin{cases} R - 1.5 & L < 5 \\ R + 1 + NUM\_SILTP_{(x,y)} & L \geq 5 \end{cases} \tag{8}$$

where $R_{dawn}$ is the adjusted distance measurement threshold $R$ at dawn. When the scene factor is less than 5, the distance measurement threshold is reduced in anticipation that the dawn line will cross this border, in order to ensure that the fog edge can still be completely detected. When either the scene factor is greater than 5, the pixel is inside fog with uniform texture, or after the dawn line has passed, then the distance measure threshold can be increased to reduce both cloud and surface false detection rates.

The method for determining $R$ at dusk is:

$$R_{dusk} = \begin{cases} 1\,\text{K} & L < 0 \\ 1.5\,\text{K} & 0 \leq L < 10 \\ 2\,\text{K} & L \geq 10 \end{cases} \tag{9}$$

where $R_{dusk}$ is the adjusted distance measurement threshold $R$ at dusk. When the scene factor is less than 0, the pixel brightness temperature difference is less than 0 (which is the area after the dusk line passes), and $R_{dusk}$ is set to 1 K. When the scene factor is greater than 10, the area lies before the dusk line crossing, and $R_{dusk}$ is set to 2 K. When the scene factor is between 0 and 10, the area is near the dusk line, and $R_{dusk}$ is set to 1.5 K.

### 2.4.4. Foreground Detection and Background Update

The adaptive distance measure threshold $R$ was used to detect the foreground and update the background according to the foreground and background determination rules of the traditional VIBE algorithm. The ST-ViBe algorithm rapidly updated the background by increasing the number of background updates and increasing the update rate. All pixels detected as background points were updated quickly. Half of the samples ($n/2$) of its background model $BM(I)_1$ and $BM(I)_2$ were randomly selected and replaced with the current values of pixels so that the background model contained more recent background values.

### 2.4.5. Traditional Cloud Removal Methods

The mid-high clouds were subsequently removed through use of the differences in the spectral and textural characteristics between mid-high clouds and fog, which we described in detail in our previous study [18]. Specifically, the ice clouds were removed first by using a threshold of TIR (10.4 μm) brightness that was lower than 230 K [31]. Then, the pixels with BTDs greater than 0 K were taken to be thin cirrus clouds and were removed in order to identify fog. Finally, the mid-high clouds were identified through their complex texture and were removed using the infrared and visible bands texture filtering method [32].

### 2.4.6. Postprocessing

Affected by atmospheric movement, clouds move fast and easily form residual shadows, which can be mistakenly detected as fog. This algorithm took the cloud detection results of remote sensing images within a certain time range and superimposed them over the fog detection results of the current moment, and the intersecting pixels could then be determined as cloud false detections and removed. In addition, for some noise points caused by the same spectrum of foreign objects, a $3 \times 3$ median window was used to denoise the detection results, and the final fog detection results were obtained.

### 2.5. Evaluation of the Method

The probability of detection (POD), false alarm ratio (FAR), and critical success index (CSI) were adopted for the validation of fog detection, as they are the most commonly used indexes for fog detection [33–35]. The higher the POD value, the better the algorithm is at capturing real fog events. The lower the value of FAR, the lower the probability that the algorithm incorrectly estimates the fog event. CSI represents the stability of the algorithm; the higher CSI value is, the more stable the algorithm is. The three indexes are described as follows:

$$\text{POD} = \frac{N_H}{N_H + N_M} \tag{10}$$

$$\text{FAR} = \frac{N_F}{N_H + N_F} \tag{11}$$

$$\text{CSI} = \frac{N_H}{N_H + N_M + N_F} \tag{12}$$

where $N_H$, $N_M$ and $N_F$ are the pixel numbers of hitting, missing, and false detections, respectively. All indicators are scaled from 0 to 1, with high POD and CSI and low FAR indicating the great performance of the algorithm.

## 3. Results and Discussion

### 3.1. Qualitative Analysis of the Time-Series Fog Detection Results of the ST-ViBe Algorithm

The ST-ViBe algorithm was used to carry out fog detection at dawn and dusk in the study area, and the results are shown in Figures 6 and 7. The results show that the algorithm can effectively extract the fog near the dawn and dusk line (yellow line in Figures 6 and 7) when brightness temperature differences of fog and surface are close. At dawn, with the passage of the dawn line, the fog area detected by the algorithm gradually increased (the area between the yellow line and black line in Figure 6b,d,f,g). Fog on the right side of the dawn line was detected in the area with a high solar zenith angle. At dusk, the detection range moved with the dusk line, and fog within a certain range on the left and right sides of the dusk line was effectively detected (the area between two black lines in Figure 7b,d,f,g), indicating that the algorithm was suitable for fog detection in areas with high solar zenith angles at dusk and when just entering the night. Some of the fog detection results at dusk had slight afterimages, which were traces of clouds moving fast.

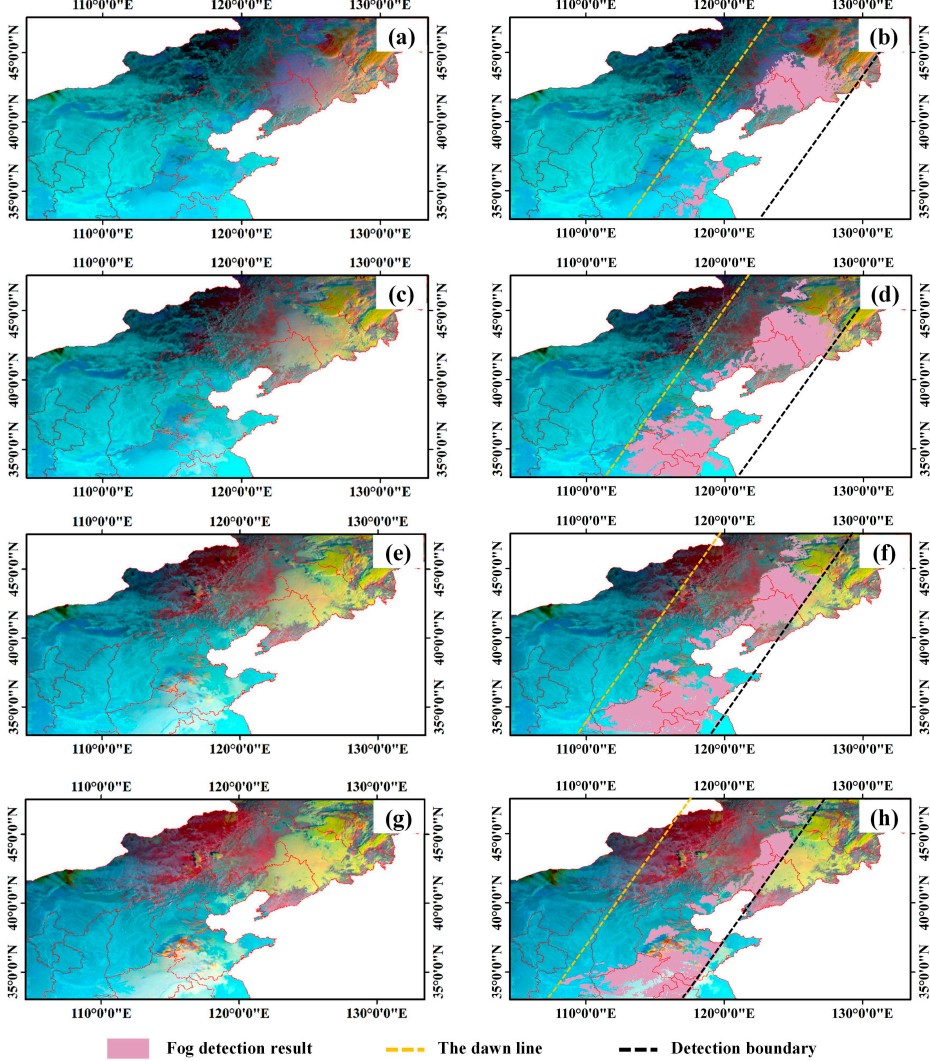

**Figure 6.** The false color images and detection results of the ST-ViBe algorithm at 7:10 (**a**,**b**); 7:30 (**c**,**d**); 7:50 (**e**,**f**); and 8:10 (**g**,**h**), on 30 November 2015. The backgrounds of the images are false color images with MIR 3.9 μm (R), TIR 8.6 μm (G), and TIR 11.2 μm (B).

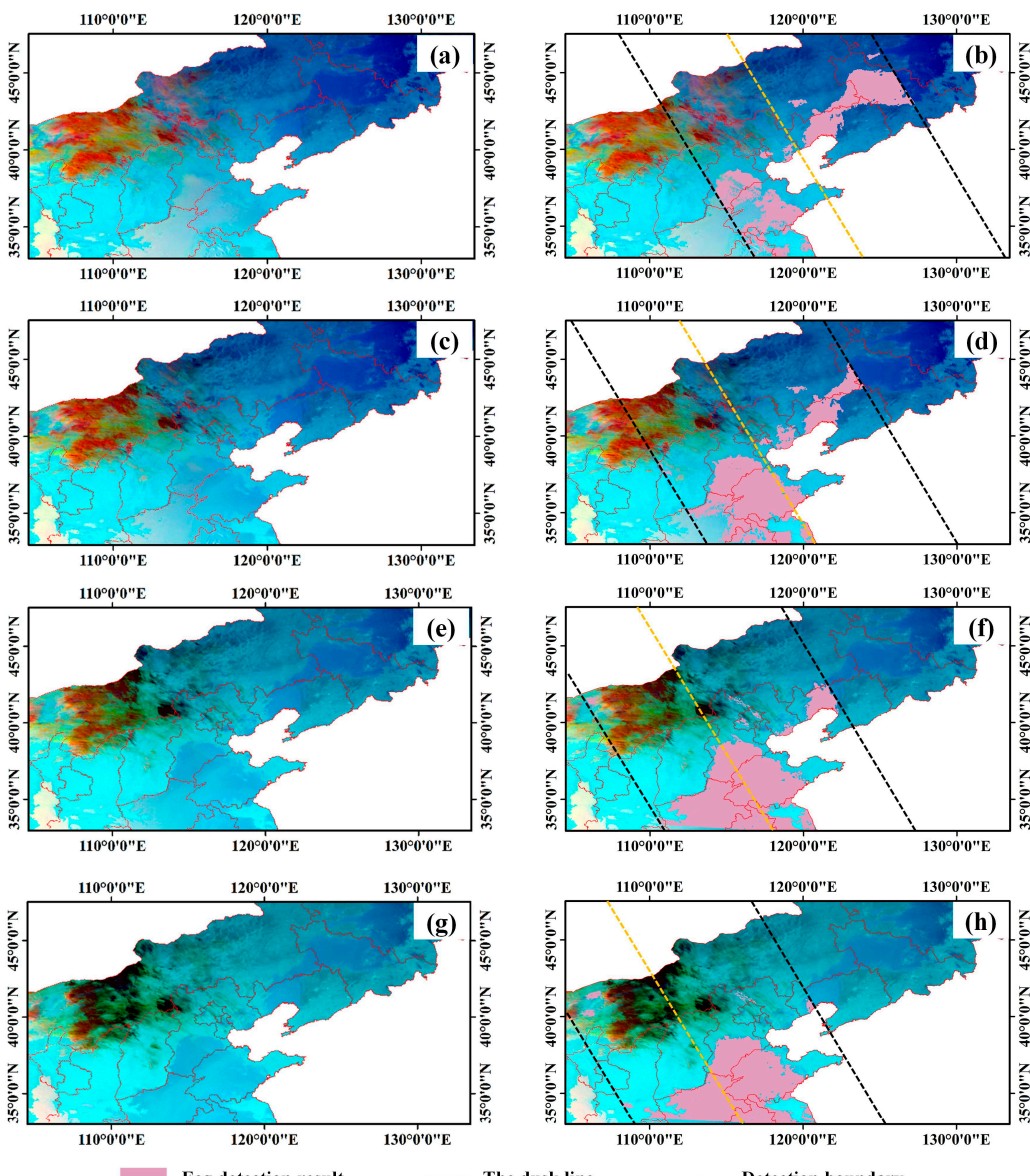

**Figure 7.** The false color images and detection results of the ST-ViBe algorithm at 16:10 (**a**,**b**); 16:30 (**c**,**d**); 16:50 (**e**,**f**); and 17:10 (**g**,**h**), on 30 November 2015. The backgrounds of the images are false color images with MIR 3.9 μm ®, TIR 8.6 μm (G), and TIR 11.2 μm (B).

The ST-ViBe algorithm has a certain difference in nighttime adaptability at dawn and dusk. The main reason is that the area before the dawn line crossing (the left sides of the dawn line in Figure 6b,d,f,g) are always in the dark, and the fog BTD changes are slight, making it difficult to carry out background detection. Although the area after the dusk line crossing is in the dark, the previous part of the area is in the daytime and has a significant change rate compared with the current pixel, which can effectively distinguish the foreground; therefore, the dusk time algorithm is suitable for the high solar zenith angles at dusk and the beginning of night.

### 3.2. Qualitative Analysis of Algorithm Fog Detection Results

To test the effectiveness of the ST-ViBe algorithm, a qualitative comparative analysis was carried out on the fog detection results of the ST-ViBe algorithm, the traditional ViBe algorithm (R = 3 K), and the improved ViBe algorithm (R = 3 K) [36]. The algorithm detection results are shown in Figures 8 and 9. The comparison results show that the ST-ViBe algorithm provides relatively complete detection of fog over a wide range at dawn

and dusk, and the interior of the fog is intact. At dawn, it is difficult for the ViBe algorithm to detect such fog near the dawn line that is consistent with the spectral information of the ground surface (Figure 8b). The detection result of the improved ViBe algorithm was similar to that of the VIBE algorithm (Figure 8c). In contrast, the ST-ViBe algorithm can achieve relatively complete detection of such fog near the dawn line with spectral characteristics similar to those of the ground surface, as well as fog in the dawn line transit area (Figure 8d). The ST-ViBe algorithm also had high fog detection accuracy at dusk, while the fog area detected by the ViBe algorithm was larger than the actual fog area, and there was also a false detection of the cloud area (Figure 9b). The improved ViBe algorithm had a better detection result; however, there was some noise and false detection (Figure 9c). The ST-ViBe algorithm can achieve relatively complete detection of fog on both sides of the dusk line, but there were a few clouds falsely detected as fog.

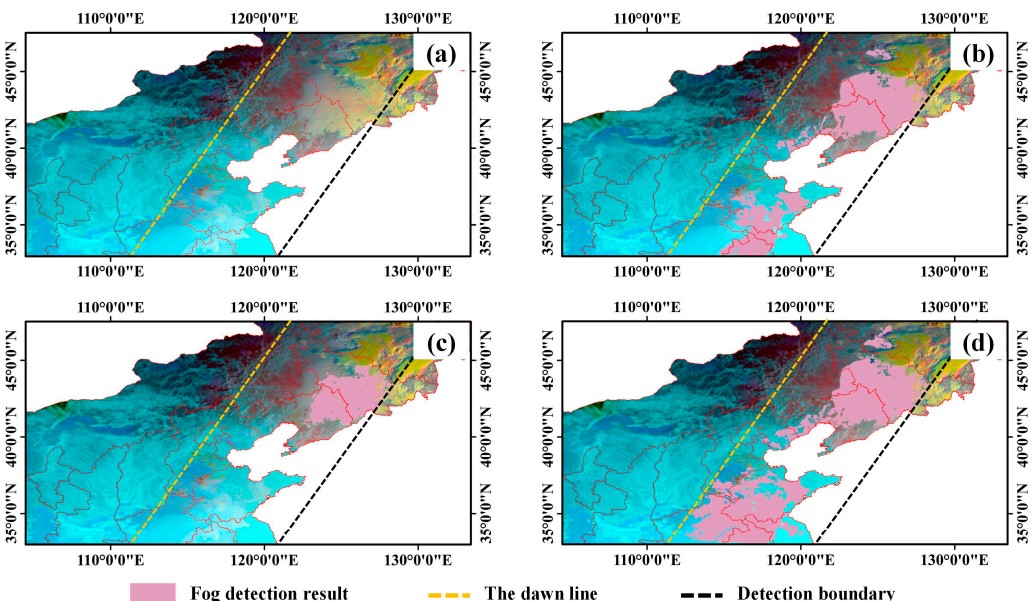

**Figure 8.** (**a**) False color image, (**b**) ViBe algorithm, (**c**) Improved ViBe algorithm, and (**d**) ST-ViBe algorithm fog detection results at 7:30 on 30 November 2015.

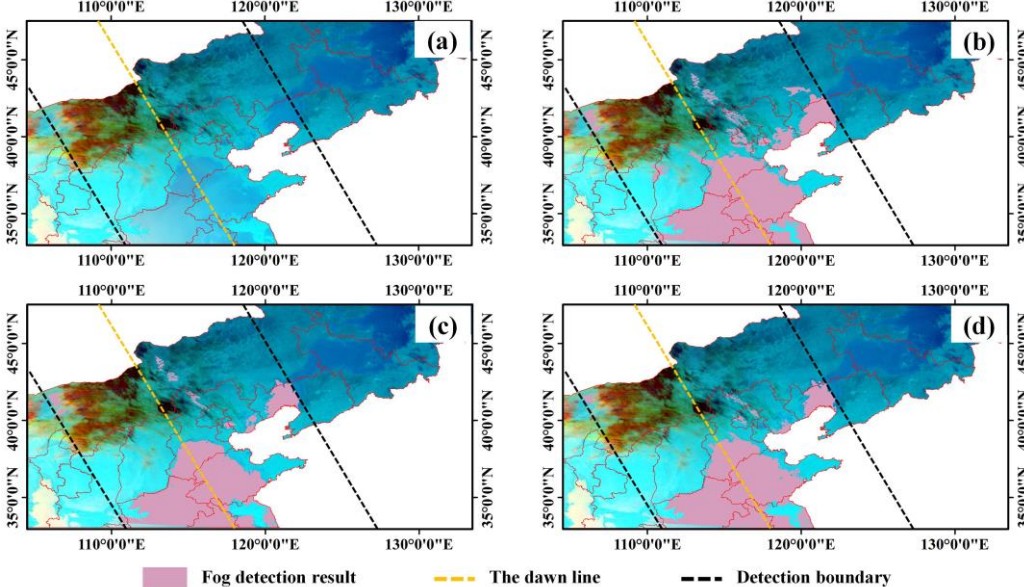

**Figure 9.** (**a**) False color image, (**b**) ViBe algorithm, (**c**) Improved ViBe algorithm, and (**d**) ST-ViBe algorithm fog detection results at 16:50 on 30 November 2015.

### 3.3. Quantitative Verification of the ST-ViBe Algorithm

The ground observation data at 8:00 and 17:00 from 27–30 November 2015, were further selected to quantitatively evaluate the accuracy of the fog detection results (Figure 10). The results showed that the ST-ViBe algorithm could effectively detect strong fog and dense fog, while haze and no fog also have good spatial consistency with the ground stations. Moreover, that dense fog has higher frequency in mountain areas (Figure 10b,c,e–h) when compared in the plain regions, the main reason being that topographic relief has an important impact on the ground wind field, temperature field, and humidity field [37]. However, there were false detections at some moments. For example, on the dawns of November 27 and November 29, the detection results in the northern part of the study area were fog (Figure 10a,c), while the ground observation results were not, because low clouds were being falsely detected as fog. In addition, the fog detection results at dusk from November 28 to 29 detected clouds as fog in the north (Figure 10f,g), mainly due to the complex cloud types and structures in Inner Mongolia and Gansu Province, which had a certain impact on the algorithm.

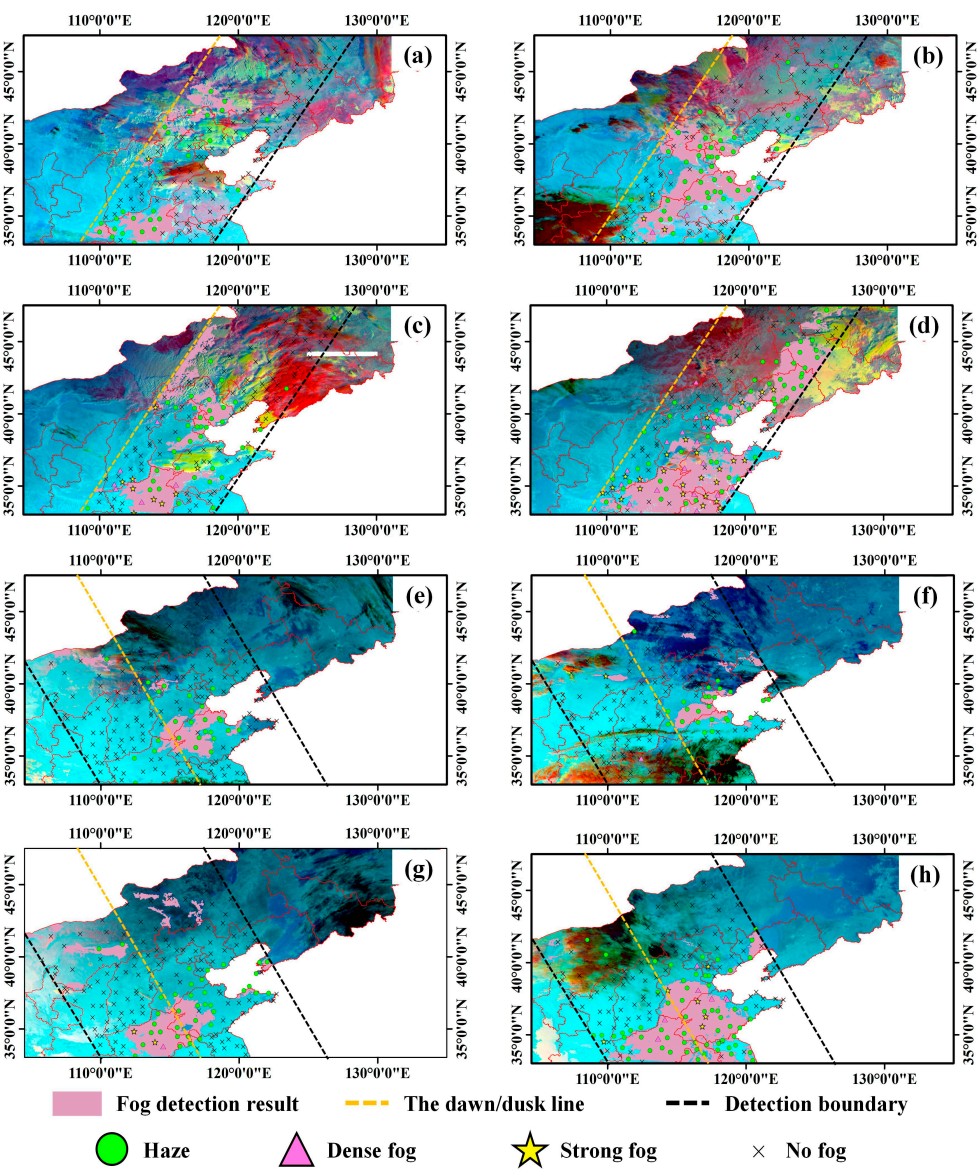

**Figure 10.** Ground observation and fog detection results based on the ST-VIBE algorithm at 8:00 (**a–d**) and 17:00 (**e–h**) from 27–30 November 2015.

Tables 1 and 2 show the statistical results of ground verification of the ST-ViBe algorithm at dawn and dusk. The average probability of fog detection by ST-ViBe in the dawn (8:00 Beijing time) was 72.9%, the average false alarm ratio was 12.7%, and the average critical success index was 66%. At dusk (17:00), the accuracy dropped slightly, with an average probability of detection of 68.9%, an average false alarm ratio of 14.5%, and an average critical success index of 61.7%. As shown in Figure 10, the missed detection points in Tables 1 and 2 were mostly haze, mainly because the haze was thin, which makes it difficult for the sensor to effectively capture its radiation information. False detection points were mostly located in the fog detection results. Because the algorithm extracted information based on spectral features, the sensor received information on the top of the target, and most of the radiation fog was surface-shaped water vapor condensation. The actual weather conditions on the ground were not only affected by water vapor but also affected by the land use type, wind speed, and terrain. There might also have been fog dissipating or ungrounded as low clouds, resulting in false detection. In the future, how to combine the above factors to further screen the satellite detection results to obtain more accurate detection results must be investigated. In addition to misdetection in the satellite-based detection results, ground observation data can also cause errors due to the recording rules. For example, when weather phenomena such as haze, clouds, rain, and snow occur at the same time as fog, the ground observation station will preferentially record rain and snow, while haze and fog are blurred, both of which affects the accuracy of the detection results.

**Table 1.** Detection accuracy at 8:00 in the study area.

| Date | Satellite Detection | Ground Observation (Fog) | Ground Observation (Nonfog) | POD | FAR | CSI |
|---|---|---|---|---|---|---|
| 27 November 2015 | Fog | 21 | 4 | 0.724 | 0.160 | 0.636 |
| | Nonfog | 8 | 138 | | | |
| 28 November 2015 | Fog | 42 | 7 | 0.689 | 0.143 | 0.618 |
| | Nonfog | 19 | 103 | | | |
| 29 November 2015 | Fog | 40 | 5 | 0.727 | 0.111 | 0.667 |
| | Nonfog | 15 | 111 | | | |
| 30 November 2015 | Fog | 86 | 9 | 0.775 | 0.095 | 0.717 |
| | Nonfog | 25 | 51 | | | |
| **Mean** | | | | 0.729 | 0.127 | 0.660 |

**Table 2.** Detection accuracy at 17:00 in the study area.

| Date | Satellite Detection | Ground Observation (Fog) | Ground Observation (Nonfog) | POD | FAR | CSI |
|---|---|---|---|---|---|---|
| 27 November 2015 | Fog | 20 | 5 | 0.741 | 0.2 | 0.625 |
| | Nonfog | 7 | 133 | | | |
| 28 November 2015 | Fog | 16 | 4 | 0.615 | 0.2 | 0.533 |
| | Nonfog | 10 | 135 | | | |
| 29 November 2015 | Fog | 29 | 2 | 0.69 | 0.065 | 0.659 |
| | Nonfog | 13 | 121 | | | |
| 30 November 2015 | Fog | 54 | 7 | 0.711 | 0.115 | 0.651 |
| | Nonfog | 22 | 82 | | | |
| **Mean** | | | | 0.689 | 0.145 | 0.617 |

This study further evaluated the performances of the algorithm over different months (Tables 3 and 4), which present universally acceptable accuracies. Validation of fog detection at dawn showed that the values of POD, FAR, and CSI were 0.725, 0.185, and 0.624 for the overall accuracies in the whole study area. The performances at dusk were relatively weak when compared with those at dawn. The indicators of POD, FAR, and CSI were 0.706, 0.336, and 0.523, respectively, over the whole study area. The main reason for this might be attributed to the calculation of validation indicators, which we have explained in our previous study in detail [18]. According to the Equations (10)–(12), the evaluation indicators were strongly determined by fog frequencies. As a result, the low frequencies of fog occurrence may generate significant uncertainties for the validation. Even with the uncertainty, however, the accuracies were acceptable when compared with most of the previous studies. Likewise, the high values of nonfog detection in Tables 3 and 4 confirm the reliability of our algorithm.

**Table 3.** Fog detection accuracy at 8:00 (Beijing Time) over different months.

| Cases | Satellite Detection | Ground Observation (Fog) | Ground Observation (Nonfog) | POD | FAR | CSI |
|---|---|---|---|---|---|---|
| 5 January 2017 | Fog | 58 | 5 | 0.744 | 0.079 | 0.699 |
| | Nonfog | 20 | 88 | | | |
| 5 February 2017 | Fog | 70 | 10 | 0.795 | 0.125 | 0.714 |
| | Nonfog | 18 | 73 | | | |
| 17 March 2017 | Fog | 25 | 6 | 0.595 | 0.194 | 0.521 |
| | Nonfog | 17 | 123 | | | |
| 6 April 2017 | Fog | 56 | 11 | 0.700 | 0.164 | 0.615 |
| | Nonfog | 24 | 80 | | | |
| 10 May 2017 | Fog | 20 | 3 | 0.769 | 0.130 | 0.690 |
| | Nonfog | 6 | 142 | | | |
| 7 June 2017 | Fog | 48 | 20 | 0.706 | 0.294 | 0.545 |
| | Nonfog | 20 | 83 | | | |
| 26 July 2017 | Fog | 30 | 6 | 0.638 | 0.167 | 0.566 |
| | Nonfog | 17 | 118 | | | |
| 31 August 2017 | Fog | 45 | 6 | 0.672 | 0.118 | 0.616 |
| | Nonfog | 22 | 97 | | | |
| 17 September 2017 | Fog | 40 | 3 | 0.889 | 0.070 | 0.833 |
| | Nonfog | 5 | 123 | | | |
| 12 October 20170 | Fog | 30 | 10 | 0.789 | 0.250 | 0.625 |
| | Nonfog | 8 | 123 | | | |
| 5 November 2017 | Fog | 34 | 12 | 0.708 | 0.261 | 0.567 |
| | Nonfog | 14 | 111 | | | |
| 20 December 2017 | Fog | 7 | 4 | 0.700 | 0.364 | 0.500 |
| | Nonfog | 3 | 157 | | | |
| | | | Mean | 0.725 | 0.185 | 0.624 |

**Table 4.** Fog detection accuracy at 17:00 (Beijing Time) over different months.

| Cases | Satellite Detection | Ground Observation (Fog) | Ground Observation (Nonfog) | POD | FAR | CSI |
|---|---|---|---|---|---|---|
| 5 January 2017 | Fog | 30 | 11 | 0.857 | 0.268 | 0.652 |
| | Nonfog | 5 | 119 | | | |
| 21 February 2017 | Fog | 8 | 7 | 0.727 | 0.467 | 0.444 |
| | Nonfog | 3 | 147 | | | |
| 13 March 2017 | Fog | 6 | 3 | 0.545 | 0.333 | 0.429 |
| | Nonfog | 5 | 151 | | | |
| 5 April 2017 | Fog | 32 | 8 | 0.914 | 0.200 | 0.744 |
| | Nonfog | 3 | 122 | | | |

**Table 4.** *Cont.*

| Cases | Satellite Detection | Ground Observation (Fog) | Ground Observation (Nonfog) | POD | FAR | CSI |
|---|---|---|---|---|---|---|
| 11 May 2017 | Fog | 7 | 2 | 0.778 | 0.222 | 0.636 |
| | Nonfog | 2 | 154 | | | |
| 7 June 2017 | Fog | 1 | 1 | 0.333 | 0.500 | 0.250 |
| | Nonfog | 2 | 161 | | | |
| 17 July 2017 | Fog | 11 | 4 | 0.786 | 0.267 | 0.611 |
| | Nonfog | 3 | 147 | | | |
| 9 August 2017 | Fog | 3 | 4 | 0.750 | 0.571 | 0.375 |
| | Nonfog | 1 | 157 | | | |
| 15 September 2017 | Fog | 5 | 1 | 0.556 | 0.167 | 0.500 |
| | Nonfog | 4 | 155 | | | |
| 5 October 2017 | Fog | 25 | 9 | 0.781 | 0.265 | 0.610 |
| | Nonfog | 7 | 124 | | | |
| 6 November 2017 | Fog | 11 | 4 | 0.846 | 0.267 | 0.647 |
| | Nonfog | 2 | 148 | | | |
| 22 December 2017 | Fog | 3 | 3 | 0.600 | 0.500 | 0.375 |
| | Nonfog | 2 | 157 | | | |
| | | | Mean | 0.706 | 0.336 | 0.523 |

## 4. Conclusions

In this study, third-generation geostationary meteorological satellite H8 data with high temporal resolution were selected, and an ST-ViBe algorithm was proposed to detect fog at dawn and dusk, with the ViBe algorithm as the framework. The results show that the algorithm had high accuracy. Compared with the ViBe algorithm and the improved ViBe algorithm, ST-ViBe can achieve relatively complete detection of fog over a wide range and in cases of light mutation and increased cloud interference. At dawn and dusk, the correct detection rates were 72.5% and 70.6%, and the false detection rates were 18.5% and 33.6%, respectively. In addition, the integrity of fog detection was good. At dawn, the algorithm can achieve relatively complete detection of the fog around the dawn line and after the dawn line transitions. At dusk, the algorithm can achieve relatively complete detection of the fog on both sides of the dusk line. Compared with the deep learning method, ST-ViBe was free from sample interference, with faster detection speeds and a more stable detection effect [17].

The disadvantage of the ST-ViBe algorithm was that it performs fog detection based on motion characteristics, which makes it easy to detect cloud and fog afterimages, especially when the cloud moves rapidly, which causes the fog detection range to be too large. The reflection characteristics of low clouds and fog are similar at dawn and dusk, and it was difficult for the algorithm to effectively separate them. In the meantime, the algorithm likely caused missed detection of haze. Future research will continue to screen fog detection results based on ground features to reduce false detection and provide an accurate remote sensing-based detection method for automatic fog detection at dawn and dusk over a large area. This study focused on the occurrence of fog, which it described in qualitative way by dividing the fog into sub-categories of strong, dense, and haze. The quantitative visibility data can be collected for improving validation in the future research.

**Author Contributions:** Conceptualization, H.M. and H.F.; methodology, H.M. and H.F.; software, Z.L. and K.J.; validation, B.J. and S.H.; formal analysis, H.M. and H.F.; investigation, H.F.; resources, H.M.; data curation, H.M., Z.L. and K.J.; writing—original draft preparation, H.M. and H.F.; writing—review and editing, H.F.; visualization, B.J. and S.H.; supervision, H.F.; project administration, H.F.; funding acquisition, H.M. and H.F. All authors have read and agreed to the published version of the manuscript.

**Funding:** This research was funded by National Natural Science Foundation of China [Grant No. 42071334, 42071378], and the Nature Science Foundation of Hunan Province [No. 2020JJ3045].

**Data Availability Statement:** The data presented in this study are available on request from the corresponding author.

**Conflicts of Interest:** The authors declare no conflict of interest.

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
