# Peer review of "A Novel ST-ViBe Algorithm for Satellite Fog Detection at Dawn and Dusk"

_remotesensing, doi:10.3390/rs15092331_

Round 1
Reviewer 1 Report
The article is devoted to the identification of fog near the terminator line using meteorological satellite data. The problem is difficult because the brightness temperature difference under fog conditions changes noticeably with solar zenith angle (Fig. 3a, b). Influence of low clouds and their motion additionally complicate the situation. To solve this problem, a novel ST-ViBe algorithm was developed, which is more flexible than the traditional ViBe algorithm. The capabilities of ST-ViBe are studied by comparison with ground-based data. Importantly, the comparison is presented in full, showing not only the advantages, but also possible weaknesses of the novel algorithm, such as a relatively low critical success index or a high false alarm ratio in certain cases.
The material has the scientific significance and it is clearly presented. My remarks in the attached file are mostly technical. I suggest that the article can be published in Remote Sensing after minor revision.

Author Response
Comment: The article is devoted to the identification of fog near the terminator line using meteorological satellite data. The problem is difficult because the brightness temperature difference under fog conditions changes noticeably with solar zenith angle (Fig. 3a, b). Influence of low clouds and their motion additionally complicate the situation. To solve this problem, a novel ST-ViBe algorithm was developed, which is more flexible than the traditional ViBe algorithm. The capabilities of ST-ViBe are studied by comparison with ground-based data. Importantly, the comparison is presented in full, showing not only the advantages, but also possible weaknesses of the novel algorithm, such as a relatively low critical success index or a high false alarm ratio in certain cases.
The material has the scientific significance and it is clearly presented. My remarks in the attached file are mostly technical. I suggest that the article can be published in Remote Sensing after minor revision:
Response: We highly appreciate you reviewed our work and are encouraged by your positive comments. We read your comments carefully and then revised/improved our manuscript strictly. Revised portions are marked in red in the revised manuscript. We hope the revised version had met your comments and could be published on Remote Sensing.
Comment 1# 13-14. ‘MIR and TIR (BTD)’ – these abbreviations should be explained as mid-infrared and thermal infrared (and in the main text as well, L. 43-44; now the explanation is in L. 109-110).
Response: Thanks very much for your valuable comments. We revised it as your suggestion. Please see line 44 of the revised manuscript.
Comment 2# 50. ‘dawn/dust’ – should be ‘dawn/dusk’. The same misspelling is in L. 56, 60, 130.
Response: We are sorry for the writing errors. We checked the writing throughout the manuscript and revised them according to your suggestions.
Comment 3# 59. ‘H8/AHI’ – the abbreviation ‘AHI’ should also be explained (‘Advanced Himawari Imager’).
Response: Thanks very much for your reminding. We revised it as your suggestion and changed it to Himawari-8/Advanced Himawari Imager (H8/AHI). Please see line 59 of the revised manuscript.
Comment 4# 61. ‘there are few relevant studies’ – it is necessary to mention these works on satellite detection of fog at high solar zenith angles. At least, own work [27] can be cited here. Besides, the inclusion of another own publication, Ran et al., 2022 (https://doi.org/10.3390/rs14174328) seems absolutely necessary, in particular, in the Discussion.
Response: Thanks very much for your reminding. We revised it as your suggestion and added references and discussions. Please see lines 61 to 62 and 456 to 457 of the revised manuscript.
Comment 5# 102. The lower right map should be minimally commented on in the caption. A reader should see from the caption what is shown on this map and when it is obtained.
Response: Thanks very much for your valuable comments. We revised it as your suggestion and described the calculation method of BTD images and shooting time in the caption. Please see lines 103 to 104 of the revised manuscript.
Comment 6# 106. ‘1.Satellite datasets’ – the numeration of items as 1, 2,… can confuse readers. Is it possible to delete these numbers or to use the forms 2.1.2.1, 2.1.2.2,..?
Response: Thanks very much for your reminding. We revised it as your suggestion changed the format (for example, 2.1.2.1). Please see line 108 of the revised manuscript.
Comment 7# 107-108. ‘Himawari-8 satellite observation data were selected for fog detection because there was fog at that moment’. The explanation of the choice is not entirely clear, the studied time range should be indicated to understand from this paragraph, whether days, months or years were analyzed.
Response: We are sorry for the ambiguous description. We revised it as your suggestion and rephrased the sentence. Please see lines 109 to 111 of the revised manuscript.
Comment 8# 133. ‘BTD is less than 0 k’ – probably ‘0 K’ (Kelvins). Please see Eq. 9 as well.
Response: Thanks very much for your reminding. We revised it as your suggestion and changes have been made to the full text. Please see line 140 of the revised manuscript.
Comment 9# 141. Yellow color shows the terminator line. This feature should be noted in the caption (for example, as in Fig. 6).
Response: Thanks very much for your valuable comments. We revised it as your suggestion and marked the yellow dotted line. Please see line 149 of the revised manuscript.
Comment 10# 159. ‘backward differential absolute value of fog’ – this combination of words is absent in the main text, but it appears in the caption. Its meaning should be explained.
Response: Thanks very much for your valuable comments. We revised it as your suggestion and explained it in the caption. The backward differential absolute value of fog means the absolute values of BTD at the current time minus BTD at the previous time. We clarified it in lines 167 to 169 of the revised manuscript.
Comment 11# 189. ‘n samples’ – ‘n’ should be in italics. Use of italics in the text should be checked.
Response: Thanks very much for your reminding. We revised it as your suggestion and changes have been made to the full text. Please see line 196 of the revised manuscript.
Comment 12# 211. ‘Buiding’ – should be ‘Building’ (in the scheme).
Response: Thanks very much for your valuable comments. We revised it as your suggestion and corrected the flow chart. Please see fig. 4(line 218) of the revised manuscript.
Comment 13# 219. ‘Figure.5’ – a dot is unnecessary after ‘Figure’. Analogous dots should be removed in the text.
Response: Thanks very much for your reminding. We revised it as your suggestion and changes have been made to the full text. Please see line 227 of the revised manuscript.
Comment 14# 221. ‘������(���)1’ – it is worth using the subscript, ‘������(���)1’.
Response: Thanks very much for your reminding. We revised it as your suggestion and changes have been made to the full text. Please see line 227 of the revised manuscript.
Comment 15# 228. What is the principle of selecting external pixels? Why, for example, I9 is labeled and the pixel between it and I16 is not?
Response: We are sorry for the ambiguous description. The core idea of LBSP encoding is to compare the similarity between the central pixel point and 16 adjacent pixels, and then encode. We have described it in the article. Please see Fig. 5(line 235) and lines 224 to 226 of the revised manuscript.
Comment 16# 237. ‘Shengcai L et al [24]’ – should be ‘Liao S. et al. [24]’.
Response: Thanks very much for your valuable comments. We revised it as your suggestion. Please see line 244 of the revised manuscript.
Comment 17# 251. ‘the solar zenith angles is larger than 90’ – should be ‘angles are’; the degree sign is needed, ‘is larger than 90°’.
Response: Thanks very much for your valuable comments. We revised it as your suggestion. Please see line 258 of the revised manuscript.
Comment 18# 252. ‘the solar zenith angles is small’ – should be correspondence in grammatical number, ‘the solar zenith angles are small’.
Response: Thanks very much for your valuable comments. We revised it as your suggestion. Please see line 259 of the revised manuscript.
Comment 19# 273. The variance is a squared characteristic, so the range should be indicated as , not as.
Response: Thanks very much for your reminding. We revised it as your suggestion and changes have been made to the full text. Please see line 279 and 280 of the revised manuscript.
Comment 20# 357. What is ‘the ViBeBgR algorithm’? It is mentioned for the first time in the text.
Response: Thanks for your comment. ViBeBgR is an improved ViBe algorithm. We have changed the abbreviation to improved ViBe and added a reference to it. Please see the line 369 of the revised manuscript.
Comment 21# 393. ‘probability of detection of fog detection’ – the first ‘detection’ is unnecessary. A similar construction is in L. 404 (‘the type of land use type’).
Response: Thanks very much for your reminding. We revised it as your suggestion. Please see line 408 and 419 of the revised manuscript.
Comment 22# 404-405. ‘There will be fog dissipating or ungrounded as low clouds’ – why is the future tense used?
Response: Thanks for your comment. We modified the tense and changed it to might. Please see line 419 of the revised manuscript.
Comment 23# 452. ‘H. M and H.F.’ – it is worth unifying the use of spaces and dots in the Author Contributions.
Response: Thanks very much for your reminding. We revised it as your suggestion and unified the format (e.g. H. M.). Please see lines 469 of the revised manuscript.
Comment 24# 461. ‘The data are not publicly available due to restrictions of privacy and ethical’ – this information about data on fog sounds enigmatically. In my opinion, it is simpler deleting this sentence (or explaining in more detail).
Response: Thanks very much for your reminding. We revised it as your suggestion and removed this sentence. Please see line 478 of the revised manuscript.
Comment 25# 472. ‘GIsci’ – should be ‘GISci’ (GIScience).
Response: Thank you for your valuable comment. We revised it as your suggestion. Please see line 489 of the revised manuscript.
Comment 26# 496-497. Last names of authors should not be in capitals. In addition, the fragment ‘& Mdash; Japan &Rsquo;s’ should be corrected.
Response: Thanks very much for your reminding. We have modified the format of the reference. Please see lines 513 to 515 of the revised manuscript.

Reviewer 2 Report
This is a review of “A novel ST-ViBe algorithm for satellite fog detection at dawn 2 and dusk” by Huiyun Ma , Zengwei Liu , Kun Jiang , Bingbo Jiang , Huihui Feng, and Shuaifeng Hu. I complement the authors for working on subject that is potentially useful and significant with a well written manuscript that has exceptionally well-done illustrations. However, a choice on visibility criteria, lack of supporting material and a conflict between satellite based values and surface observation means that I must ask for significant revisions which are explained in the following paragraphs.
A.Visibility Criteria Needs Correction
Lines 124-128 states “The fog occurrences were identified by the indicators of visibility and relative humidity, with the criteria of visibility of less than 10 km and relative humidity larger than 90% was marked as fog. The data were labelled strong, dense, and haze with visibility from 0 m to 200 m, 200 m to 500 m, and 500 m to 10000 m”. I strongly object to this large range of visibility which covers a wide range of possibly unrelated weather conditions and goes against the well accepted, international view that fog is defined as associated with visibilities less than or equal to 1 km (https://glossary.ametsoc.org/wiki/Fog, or any textbook). The 1 km fog criteria is well based on how visibility is related to physical aspects as particles (for example, Gultepe et al. 2009). It is also well connected to critical operational applications such as aircraft cannot land, large ships cannot dock and only the foolhardy would drive in it. This is not so for the upper portions of the authors’ 500 m to 10000 m range. In fact, 4 – 5 km visibility with greater than 90 % humidity could be a nice day in Atlantic Canada or coastal California.
It would make more sense to define visibilities in narrower ranges that have some relationship to the boundary layer atmosphere. This could be something as very dense fog (< 200 m), dense fog (< 500 m), fog (<= 1000 m), near fog or dense mist (1-2 km), mist (2- 4.8 km) and thin mist ( 4.8- 10 km). The latter three are more subjective and vary according to the observation code system being used. Nevertheless, the authors’ visibility criteria are unacceptable. They need to redo their calculations with reasonable criteria that reflect the observations in China and elsewhere. For this reason alone, I am recommending major revisions.
B. Lack of supporting material
This manuscript would have much greater impact if there was more supporting material on direct comparisons between the surface observations and remote observations. This should include scatter diagrams of the surface station visibilities and the satellite indications. If the visibility criteria was corrected to something as was suggested in paragraph A with a more realistic range of criteria, then going from dense fog to good visibility should grade across the scatter diagrams if the system works as well as the authors suggest.
C. Conflict between satellite observations and surface station observations
I need see a more detailed presentation on what is occurring over a limited area, with a more specific comparison of surface observations with the satellite fog detection values. For example, in Figure 10, especially 10g and 10h, in the lower, left of center portion, to the west of the Yellow Sea, there is a large area of satellite detected fog (purple) with surface observations of haze, strong fog mixed in with a few others. How can this be? This mixing together of observations occurs in other frames. This is inconsistent and likely a result of inappropriate and insensitive criteria for the categories used here. The alternative is that the system proposed in this manuscript is not successful. This must be addressed with specific comparisons backed up with statistics over areas/conditions that are not so broad as to mask inconsistencies.
C. Some Smaller Items.
L100: Figure 1. BTD should be explained in the figure caption. Otherwise, this is a very well done, informative and attractive figure. Even the smaller details are apparent.
L124 – 126: “The fog occurrences were identified by the indicators of visibility and relative humidity, with the criteria of visibility of less than 10 km and relative humidity larger than 90% was marked as fog”. Visibility of less than 10 km in itself is not fog and is seriously misleading to call it fog.
L126: Why is humidity greater than 90 % included in the definition?
L137-140: & Figure 2. The dash-dot line and the rectangles should be better explained in the figure caption. It is hard for me to see anything special about the red and green rectangle. Why and what are the dark areas under the blue rectangle?
L158 and Figure 3 with caption. Dashed line, fog, surface needs explanation.
L347 and Figure 6 caption. I do not understand how fog can be shown on both sides of the “detection boundary”
L390 and Figure 10: In Fig. 10b, 10c, 10e, 10f, 10g, and 10h, “haze” reports are over the “dense fog” areas which is inconsistent. The remote detection system has failed in this case. The other possibility is that the fog criteria are to broad.
L413 and all Tables. The fog criteria is too insensitive. More realistic criteria would probably significantly reduce the “success” of this tab le.
Gultepe, I., and Coauthors, 2009: The Fog Remote Sensing and Modeling Field Project. Bull. Amer. Meteor. Soc., 90, 341–360, https://doi.org/10.1175/2008BAMS2354.1.
Author Response
Comment:This is a review of “A novel ST-ViBe algorithm for satellite fog detection at dawn 2 and dusk” by Huiyun Ma , Zengwei Liu , Kun Jiang , Bingbo Jiang , Huihui Feng, and Shuaifeng Hu. I complement the authors for working on subject that is potentially useful and significant with a well written manuscript that has exceptionally well-done illustrations. However, a choice on visibility criteria, lack of supporting material and a conflict between satellite based values and surface observation means that I must ask for significant revisions which are explained in the following paragraphs.
Response: We are very grateful to your comments for the manuscript. According with your advice, we improved our work throughout the manuscript. Revised portions are marked in red in the revised manuscript. We sincerely hope the revised manuscript had met your comments and will be finally acceptable to be published on Remote Sensing.
Comment A: Visibility Criteria Needs Correction
Lines 124-128 states “The fog occurrences were identified by the indicators of visibility and relative humidity, with the criteria of visibility of less than 10 km and relative humidity larger than 90% was marked as fog. The data were labelled strong, dense, and haze with visibility from 0 m to 200 m, 200 m to 500 m, and 500 m to 10000 m”. I strongly object to this large range of visibility which covers a wide range of possibly unrelated weather conditions and goes against the well accepted, international view that fog is defined as associated with visibilities less than or equal to 1 km (https://glossary.ametsoc.org/wiki/Fog, or any textbook). The 1 km fog criteria is well based on how visibility is related to physical aspects as particles (for example, Gultepe et al. 2009). It is also well connected to critical operational applications such as aircraft cannot land, large ships cannot dock and only the foolhardy would drive in it. This is not so for the upper portions of the authors’ 500 m to 10000 m range. In fact, 4–5 km visibility with greater than 90% humidity could be a nice day in Atlantic Canada or coastal California.
It would make more sense to define visibilities in narrower ranges that have some relationship to the boundary layer atmosphere. This could be something as very dense fog (<200 m), dense fog (<500 m), fog (<=1000 m), near fog or dense mist (1-2km), mist (2-4.8 km) and thin mist (4.8-10 km). The latter three are more subjective and vary according to the observation code system being used. Nevertheless, the authors’ visibility criteria are unacceptable. They need to redo their calculations with reasonable criteria that reflect the observations in China and elsewhere. For this reason alone, I am recommending major revisions.
Response: We are sorry for the ambiguous description. Due to our oversight, 90% humidity is a redundant and meaningless description that has been removed in the paper. We apologize for any inconvenience this may cause.
Fog is usually defined as the weather condition with visibility is less than 1 km, which can be further divided into sub-categories of strong (visibility is between 0 m to 200 m), dense (200 m to 500 m), and haze (500 m to 1000 m) (Dutta, D.; Chaudhuri, S. Nowcasting Visibility during Wintertime Fog over the Airport of a Metropolis of India: De-cision Tree Algorithm and Artificial Neural Network Approach. Natural Hazards 2015, 75, 1349–1368, doi:10.1007/s11069-014-1388-9). However, the definition of haze varies at different regions. According to the Grade of Fog Forecast promulgated by China Meteorological Administration (CMA) (GB/T 27964-2011 GB/T), the haze is defined as the horizontal visibility ranges from 1.0 to 10.0 km. By considering the criterions above, this study identified the haze as the ground station is coded as 40-49 (fog occurs during observation) or 10 (haze occurs during observation) with the visibility is between 500 m to 10 km. We explained it in lines 127 to 135 of the revised manuscript.
Comment B: Lack of supporting material
This manuscript would have much greater impact if there was more supporting material on direct comparisons between the surface observations and remote observations. This should include scatter diagrams of the surface station visibilities and the satellite indications. If the visibility criteria was corrected to something as was suggested in paragraph A with a more realistic range of criteria, then going from dense fog to good visibility should grade across the scatter diagrams if the system works as well as the authors suggest.
Response: Thanks very much for your valuable comments. We absolutely agree you that it supports the direct and objective validation through the scatter diagram between surface station visibilities and remote observations. However, the ground observations used in this study did not recorded the quantitative data of the visibility, leaving the great gap for mapping the scatter diagrams with satellite indications. Furthermore, this study focused on the occurrence of fog, which described in qualitative way by dividing the fog into sub-categories of strong, dense, and haze. The quantitative visibility data would collected for improving the validation in the future researches. We explained it in lines 466 to 468 of the revised manuscript.
Comment C: Conflict between satellite observations and surface station observations
I need see a more detailed presentation on what is occurring over a limited area, with a more specific comparison of surface observations with the satellite fog detection values. For example, in Figure 10, especially 10g and 10h, in the lower, left of center portion, to the west of the Yellow Sea, there is a large area of satellite detected fog (purple) with surface observations of haze, strong fog mixed in with a few others. How can this be? This mixing together of observations occurs in other frames. This is inconsistent and likely a result of inappropriate and insensitive criteria for the categories used here. The alternative is that the system proposed in this manuscript is not successful. This must be addressed with specific comparisons backed up with statistics over areas/conditions that are not so broad as to mask inconsistencies.
Response: Thanks very much for your valuable comments. The dense fog has higher frequency in mountain areas (Fig.10b, 10c, 10e, 10f, 10g, and 10h) when compared in the plain regions, the main reason was that topographic relief has an important impact on the ground wind field, temperature field, and humidity field. (Luo, J.; Zhou, J.; Liu, J.; Lin, L. Analysis of the Heavy Mist Characteristics in Different Geographical Conditions in the Southwest of Hubei Province. Plateau and Mountain Meteorology Research 2011, 31, 51–58, doi: j.issn.1674-2184-2011.04.009) This study provides a certain research basis for fog area detection. We will further research the relation of the surface station visibilities and the satellite indications in order to provide information for fog level forecasting. We explained it in lines 394 to 396 and 466 to 468 of the revised manuscript.
Comment D: Some Smaller Items.
Comment 1# L100 Figure 1. BTD should be explained in the figure caption. Otherwise, this is a very well done, informative and attractive figure. Even the smaller details are apparent.
Response: Thanks very much for your valuable comments. We revised it as your suggestion and described the shooting time and calculation method of BTD images. Please see lines 103 to 104 of the revised manuscript.
Comment 2# L124 – 126: “The fog occurrences were identified by the indicators of visibility and relative humidity, with the criteria of visibility of less than 10 km and relative humidity larger than 90% was marked as fog”. Visibility of less than 10 km in itself is not fog and is seriously misleading to call it fog.
Response: Thanks very much for your valuable comment. We clarified the definition of fog in the revised manuscript. Specifically, fog is usually defined as the weather condition with visibility is less than 1 km, which can be further divided into sub-categories of strong (visibility is between 0 m to 200 m), dense (200 m to 500 m), and haze (500 m to 1000 m) (Dutta, D.; Chaudhuri, S. Nowcasting Visibility during Wintertime Fog over the Airport of a Metropolis of India: De-cision Tree Algorithm and Artificial Neural Network Approach. Natural Hazards 2015, 75, 1349–1368, doi:10.1007/s11069-014-1388-9). However, the definition of haze varies at different regions. According to the Grade of Fog Forecast promulgated by China Meteorological Administration (CMA) (GB/T 27964-2011 GB/T), the haze is defined as the horizontal visibility ranges from 1.0 to 10.0 km. By considering the criterions above, this study identified the haze as the ground station is coded as 40-49 (fog occurs during observation) or 10 (haze occurs during observation) with the visibility is between 500 m to 10 km. We clarified it in lines 127 to 135 of the revised manuscript.
Comment 3# L126: Why is humidity greater than 90 % included in the definition?
Response: Thanks very much for your valuable comment. We are sorry for the ambiguous description. We removed it in the revised manuscript. Please see lines 127 to 135 of the revised manuscript.
Comment 4# L137-140: & Figure 2.The dash-dot line and the rectangles should be better explained in the figure caption. It is hard for me to see anything special about the red and green rectangle. Why and what are the dark areas under the blue rectangle?
Response: Thanks for your valuable comment. We revised it as your suggestion and modified Fig. 2. We added a description of these rectangles and dashed lines. Please see Fig. 2 (line 149) of the revised manuscript.
Comment 5# L158 and Figure 3 with caption. Dashed line, fog, surface needs explanation.
Response: Thanks very much for your valuable comment. We revised it as your suggestion and modified Fig. 3. For ease of reading, we have added yellow dotted lines in Fig. 3c and Fig. 3d, which represent T=1.5 and T=0.6. We also described the fog and surface sources and the calculation principle of the backward differential absolute value of BTD in the figure caption. Please see lines 167 to 169 of the revised manuscript.
Comment 6# L347 and Figure 6 caption. I do not understand how fog can be shown on both sides of the “detection boundary”
Response: Thanks very much for your valuable comment. We have eliminated the detection results outside the boundary and re-mapped them. Please see Fig. 6 (line 358) to Fig. 10 (line 404) of the revised manuscript.
Comment 7# L390 and Figure 10: In Fig. 10b, 10c, 10e, 10f, 10g, and 10h, “haze” reports are over the “dense fog” areas which is inconsistent. The remote detection system has failed in this case. The other possibility is that the fog criteria are to broad.
Response: Thanks very much for your valuable comment. The dense fog has higher frequency in mountain areas (Fig.10b, 10c, 10e, 10f, 10g, and 10h) when compared in the plain regions, the main reason was that topographic relief has an important impact on the ground wind field, temperature field, and humidity field. (Luo, J.; Zhou, J.; Liu, J.; Lin, L. Analysis of the Heavy Mist Characteristics in Different Geographical Conditions in the Southwest of Hubei Province. Plateau and Mountain Meteorology Research 2011, 31, 51–58, doi: j.issn.1674-2184-2011.04.009) We explained it in lines 394 to 396 of the revised manuscript.
Comment 8# L413 and all Tables. The fog criteria is too insensitive. More realistic criteria would probably significantly reduce the “success” of this table.
Response: Thanks very much for your valuable comment. We are sorry for the ambiguous description. We have given a more detailed description of the discrimination rules of haze and explained why haze is defined as 500m to 10KM. This is mainly based on the China Meteorological Administration issued the Grade of Fog Forecast (GB/T 27964-2011 GB/T). We explained it in lines 127 to 135 of the revised manuscript.

Reviewer 3 Report
Fog events are harmful to transportation over land, ocean and air, but they are still difficult to predict and detect. The remote sensing detection of fog was developed in recent decade, but its quality and accuracy are still low. The new method presented in this article show a promising result. In this sense, this article is valuable and worthful to the fog research and forecast community. However, there are some issues and questions should be clarified before it can be accepted for publishing.
1. Page 3, line 124~127: " .... with the criteria of visibility of less than 10km ...." According to the definition of fog (see AMS or WMO documents), its upper limit is 1000m (1.0 km). So defining visibility <10km as fog is totally wrong! Hope this is a typo, otherwise the base of this work is wrong and could be rejected directly. In addition, the line 127 denotes that the visibility range 500m ~ 10000m is marked as haze. This is also confusing since 500m ~ 1000m has been defined as fog by WMO. In other words, the cases with visibility > 500m were excluded since haze is not fog. This implies that the detection methods suggested in this article is just for strong fog or dense fog, not for light fog (500m ~ 1000m)? Please clarify this question.
2. Page 10, line 317, suggest the title of 2.5 "Result Verification" to "Verification" or "Evaluation of the Method". Please also provide a reference for the verification scores used (Eq. 10,11, and 12). These 3 traditional and well-known scores are so called "category" verification method. But many other readers may not familiar with. So please provide a reference and some background, why use category scores to evaluate the new method
3. Page 12, line 358 and in the conclusion: "... the ST-ViBe algorithm can completely detect fog in a wide range at dawn and dusk ...." Here "completely" is not be a appropriate word and may cause misleading. The word "completely" implies that the method is perfect. But no method can be perfect in science. The results presented in Table 1,2,and 3 already showed the POD values are generally around 0.7 ~ 0.8, far from the perfect value 100% . Otherwise please give further explanation if still use the word "completely".
4. Page 18 line 448 change "easily" to "likely"
5. Page 18 line 448 "light fog". Please explain what its definition is. As I know, in different countries light fog has different definitions.
Author Response
Comment: Fog events are harmful to transportation over land, ocean and air, but they are still difficult to predict and detect. The remote sensing detection of fog was developed in recent decade, but its quality and accuracy are still low. The new method presented in this article show a promising result. In this sense, this article is valuable and worthful to the fog research and forecast community. However, there are some issues and questions should be clarified before it can be accepted for publishing.
Response: Thank you very much for your comments and professional advice. We read your comments carefully and then revised/improved our manuscript strictly. Revised portions are marked in red in the revised manuscript. We hope the revised version had met your comments and could be published on Remote Sensing.
Comment 1#: Page 3, line 124~127: " .... with the criteria of visibility of less than 10km ...." According to the definition of fog (see AMS or WMO documents), its upper limit is 1000m (1.0 km). So defining visibility <10km as fog is totally wrong! Hope this is a typo, otherwise the base of this work is wrong and could be rejected directly. In addition, the line 127 denotes that the visibility range 500m ~ 10000m is marked as haze. This is also confusing since 500m ~ 1000m has been defined as fog by WMO. In other words, the cases with visibility > 500m were excluded since haze is not fog. This implies that the detection methods suggested in this article is just for strong fog or dense fog, not for light fog (500m ~ 1000m)? Please clarify this question.
Response: Thanks for your valuable comment. Fog is usually defined as the weather condition with visibility is less than 1 km, which can be further divided into sub-categories of strong (visibility is between 0 m to 200 m), dense (200 m to 500 m), and haze (500 m to 1000 m) (Dutta, D.; Chaudhuri, S. Nowcasting Visibility during Wintertime Fog over the Airport of a Metropolis of India: De-cision Tree Algorithm and Artificial Neural Network Approach. Natural Hazards 2015, 75, 1349–1368, doi:10.1007/s11069-014-1388-9). However, the definition of haze varies at different regions. According to the Grade of Fog Forecast promulgated by China Meteorological Administration (CMA) (GB/T 27964-2011 GB/T), the haze is defined as the horizontal visibility ranges from 1.0 to 10.0 km. By considering the criterions above, this study identified the haze as the ground station is coded as 40-49 (fog occurs during observation) or 10 (haze occurs during observation) with the visibility is between 500 m to 10 km. We explained it in lines 127 to 135 of the revised manuscript.
Comment 2#: Page 10, line 317, suggest the title of 2.5 "Result Verification" to "Verification" or "Evaluation of the Method". Please also provide a reference for the verification scores used (Eq. 10,11, and 12). These 3 traditional and well-known scores are so called "category" verification method. But many other readers may not familiar with. So please provide a reference and some background, why use category scores to evaluate the new method
Response: Thanks very much for your valuable comments. We revised it as your suggestion and changed the title to "Evaluation of the Method". Please see line 324 of the revised manuscript.
The POD, FAR and CSI are the most commonly used indexes for fog detection. The higher the POD value, the better the algorithm is at capturing real fog events. The lower the value of FAR, the lower the probability that the algorithm incorrectly estimates the fog event. CSI represents the stability of the algorithm, the higher CSI value is, the more stable the algorithm is. We have described these three indexes and added relevant references. Please see lines 326 to 330 of the revised manuscript.
Comment 3#: Page 12, line 358 and in the conclusion: "... the ST-ViBe algorithm can completely detect fog in a wide range at dawn and dusk ...." Here "completely" is not be a appropriate word and may cause misleading. The word "completely" implies that the method is perfect. But no method can be perfect in science. The results presented in Table 1,2, and 3 already showed the POD values are generally around 0.7 ~ 0.8, far from the perfect value 100% . Otherwise please give further explanation if still use the word "completely".
Response: Thanks very much for your reminding. We revised it as your suggestion and modified completely to relatively intact. Please see line 371 of the revised manuscript.
Comment 4#: Page 18 line 448 change "easily" to "likely"
Response: Thanks very much for your valuable comments. We revised it as your suggestion. Please see line 463 of the revised manuscript.
Comment 5#: Page 18 line 448 "light fog". Please explain what its definition is. As I know, in different countries light fog has different definitions.
Response: We are sorry for the ambiguous description. The light fog is haze. We changed it as haze. Please see line 463 of the revised manuscript.

Round 2
Reviewer 3 Report
The authors have answered all of my questions and clarified all issues.
I have no further questions for this work and recommend to accept the
manuscript in current version